# Development and validation of a nomogram for predicting lymph node metastasis in ductal carcinoma in situ with microinvasion: A SEER population-based study

**Kaijun Zhu**[1☉], **Yuan Sui**[1☉], **Mingliao Zhu**[1], **Yuan Gao**[2], **Ying Yuan**[1], **Pujian Sun**[1], **Liwei Meng**[2]*, **Jiangfeng Dai**[3], **Zhian Li**[3]*

1 School of Medicine, Shaoxing University, Shaoxing, Zhejiang, P. R. China, 2 Department of Breast and Thyroid Surgery, Shaoxing People's Hospital, The First Affiliated Hospital of Shaoxing University, Shaoxing, Zhejiang Province, People's Republic of China, 3 Department of Oncological Surgery, Shaoxing Second Hospital, Shaoxing, Zhejiang, China

☉ These authors contributed equally to this work.
* menglw@usx.edu.cn (LM); Woshilizhian@sina.com (ZL)

**Data Availability Statement:** All relevant data are within the manuscript and its Supporting Information files.

## Abstract

### Background

Ductal carcinoma in situ with microinvasion (DCIS-MI) is a special type of breast cancer. It is an invasive lesion less than 1.0 mm in size related to simple ductal carcinoma in situ (DCIS). Lymph node metastasis (LNM) in DCIS-MI often indicates a poor prognosis. Therefore, the management of lymph nodes plays a vital role in the treatment strategy of DCIS-MI. Since DCIS-MI is often diagnosed by postoperative paraffin section and immunohistochemical detection, to obtain the best clinical benefits for such patients, we aim to establish and verify a nomogram to predict the possibility of lymph node metastasis in DCIS-MI patients and help preoperative or intraoperative clinical decision-making.

### Methods

A retrospective analysis of patients with DCIS-MI in the Surveillance, Epidemiology, and End Results (SEER) database from 2010 to 2019 was performed. The study cohort was randomly divided into a training cohort and a validation cohort at a ratio of 7:3. The risk factors were determined by univariate and multivariate logistic regression analyses in the training cohort, and a nomogram was constructed. The receiver operating characteristic (ROC) curve, calibration curve, and decision curve analysis (DCA) were used to evaluate the nomogram in the training set and validation set. An independent data cohort was obtained from the Shanghai Jiao Tong University Breast Cancer Database (SJTU-BCDB) for external validation.

### Results

This study included 3951 female patients from SEER with DCIS-MI, including 244 patients with regional lymph node metastasis, accounting for 6.18% of the total. An independent test

**Funding:** This work was supported by Nonprofit Science and Technology Projects of Zhejiang Medical and Health (No. 2021KY369).

**Competing interests:** The authors declare that the research was conducted in the absence of any commercial or financial relationships that could be construed as a potential conflict of interest.

set of 323 patients from SJTU-BCDB was used for external validation. According to the multifactorial logistic regression analysis results, age at diagnosis, ethnicity, grade, and surgical modality were included in the prediction model. The areas under the ROC curves (AUCs) were 0.739 (95% CI: 0.702~0.775), 0.732 (95% CI: 0.675~0.788), and 0.707 (95%CI: 0.607–0.807) in the training, validation and external test groups, suggesting that the column line graphs had excellent differentiation. The calibration curves slope was close to 1, and the model's predicted values were in good agreement with the actual values. The DCA curves showed good clinical utility.

## Conclusion

In this study, we constructed accurate and practical columnar maps with some clinical benefit to predict the likelihood of lymph node metastasis in patients with postoperatively diagnosed DCIS-MI and provide a reference value for specifying treatment strategies.

## Introduction

Ductal carcinoma in situ with microinvasion (DCIS-MI) is a specific type of cancer that accounts for approximately 1% of new breast cancer cases and is growing at an annual rate of 2.1% [1–3]. In 1997, the American Joint Committee on Cancer (AJCC) introduced the concept of microinvasive carcinoma into the staging system for the first time, and in the 8th edition of the AJCC staging system, DCIS-MI was clearly defined as breast carcinoma with invasive tumor lesions less than 1.0 mm in maximum diameter [4].

Several studies have reported the clinical and pathological features of DCIS-MI and found that lymph node metastasis (LNM) can occur in this group of patients and is significantly associated with a poor prognosis, with one study noting a 2.1% lymph node metastasis rate in DCIS-MI patients and a significantly lower 5-year survival rate in lymph node-positive DCIS-MI patients than in the lymph node-negative subgroup [5–7]. Therefore, the management of regional lymph nodes is crucial in the treatment of DCIS-MI patients.

However, because the foci of infiltration in microinvasive breast cancer are microscopic and easily overlooked by preoperative puncture biopsy or intraoperative frozen section examination [8, 9], it is often diagnosed as DCIS before or during the operation. Nevertheless, after the operation, it is interpreted as DCIS-MI from a continuous section of the extracted mass. Because some patients with DCIS-MI do not undergo axillary dissection or sentinel lymph node biopsy during the operation, it is difficult to make clinical decisions on whether these patients need further axillary surgery. Therefore, this study used an extensive sample database to retrospectively analyze the risk factors for lymph node metastasis in patients with microinvasive breast cancer and established a practical, sensitive, and feasible prediction model to provide the most significant clinical benefits for such patients.

## Materials and methods

### Data collection and patient selection

Through the largest publicly available cancer database—the Surveillance, Epidemiology, and End Results (SEER) database (https://seer.cancer.gov/)—the data of patients with microinvasive breast cancer from 2010 to 2019 were obtained. The inclusion criteria were as follows: (1) female patients, (2) pathological diagnosis of microinvasive breast cancer (T1mic), and (3)

breast cancer was the first and only primary tumor at the time of initial diagnosis after inclusion. We also used the following exclusion criteria;(1) patients with other cancerous diseases; (2) patients with preoperative neoadjuvant chemotherapy or preoperative radiotherapy; (3) patients with unknown estrogen receptor (ER) status; (4) patients with unknown progesterone receptor (PR) status; (5) patients with unknown human epidermal growth factor receptor 2 (HER-2) status; and (6) patients with unknown tumor grade information. Using the above criteria for screening, we finally included 3951 patients in this study. The flowchart of patients with DCIS-MI selection is shown in Fig 1.

An external test group of 323 patients diagnosed with DCIS-MI from the Shanghai Jiao Tong University Breast Cancer Database (SJTU-BCDB) was used to further validate the constructed nomogram. The inclusion and exclusion criteria were the same as the training set. We were granted permission to access relevant data in the SEER and SJTU-BCDB databases for this study and we have the right to access the information that could identify individual participants during or after data collection. This research was therefore exempted from review by the institutional review board of the First Affiliated Hospital of Shaoxing University.

## Incorporate variables

The following clinicopathological factors obtained from the SEER and SJTU-BCDB databases were included in the study as variables, and the variables were grouped. The variables included

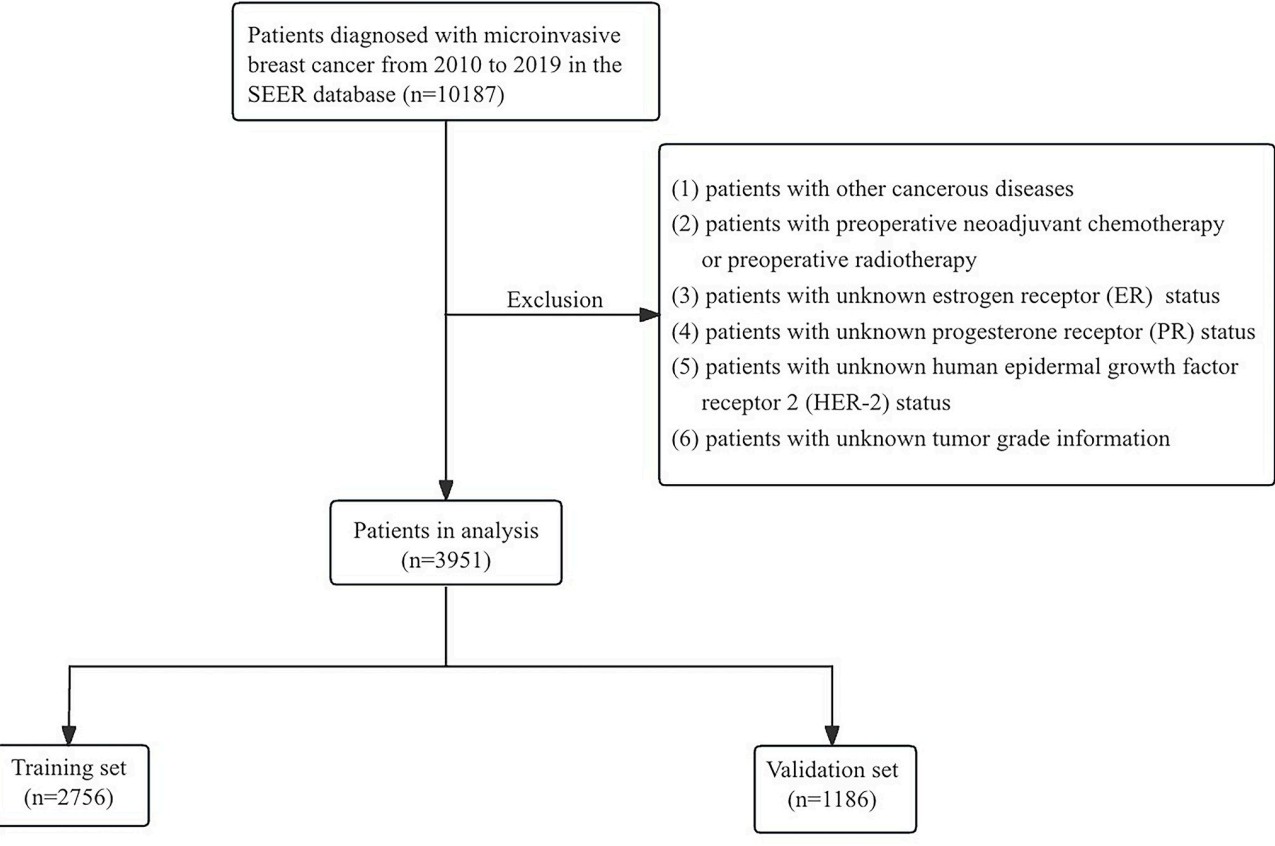

**Fig 1. The flowchart of the selection for patients with DCIS-MI in SEER database.**

age at diagnosis, race, grade, laterality, location, ER status, PR status, HER-2 status, surgical procedure, and regional lymph node status.

## Statistical analysis

In this study, all the included patients from the SEER database were divided into a training set (70%) and a validation set (30%) by a random sampling method. An independent data group was used for external validation. Pearson's chi-square or Fisher's exact test showed no significant difference between the training and validation sets. Univariate and multivariate logistic regression were used to determine the factors related to lymph node metastasis in the training set. Variables with P < 0.05 were selected to construct a nomogram for prediction, and the data in the validation set were used for internal validation. The accuracy of the nomogram was evaluated by the receiver operating characteristic (ROC) curve, and the area under the curve (AUC) was calculated. The Hosmer–Lemeshow goodness of fit was used to draw the calibration curve to illustrate the correlation between the probability of lymph node metastasis predicted by the nomogram and the actual probability. Decision curve analysis (DCA) was used to evaluate the clinical effectiveness of the prediction model. All statistical analyses in this study were performed using SPSS 24.0 (IBM Corp, Armonk, NY, USA) and R software (version 4.1.2, http://www.R-project.org). A P < 0.05 (both sides) was considered statistically significant.

## Results

### Basic characteristics of the research cohort

In this study, we included 3951 female DCIS-MI patients diagnosed from 2010 to 2019 and 244 patients who had regional lymph node metastasis, accounting for 6.18%. Table 1 summarized the clinicopathological features of 3951 patients in detail. In this study cohort, the median age of the patients was 59 years old, most of the primary sites were the outer upper quadrant (35.3%), and lesion differentiation grade IV only accounted for 0.9% of the total patients. The positive rate of ER was 76.7%, the positive rate of PR was 62.4%, and the positive rate of HER-2 was 30.3%. In terms of treatment, 58.9% of patients underwent breast-conserving surgery, 40.3% experienced a total mastectomy, and only 0.7% did not undergo surgery or the surgical method was unknown. In this study, patients were divided into a training set (n = 2765) and a validation set (n = 1186) by the random split sample method, and there was no significant difference between each variable in the training set and validation set. Meanwhile,323 patients were obtained from the SJTU-BCDB database as the external validation group, and the detailed clinicopathological features are presented in Table 2.

### Risk factors for regional lymph node metastasis in DCIS-MI patients

To determine the potential risk factors for lymph node metastasis, binary logistic regression analysis was performed on age at diagnosis, race, laterality, localization, histological grade, ER, PR, HER-2 status, and surgical procedure. Univariate logistic regression analysis showed that all included variables were significantly correlated with LNM (P < 0.0001). Therefore, these variables were added to the multivariate analysis. The results showed that age at diagnosis, race, grade, and surgical procedure were closely related to the occurrence of LNM. Among them, black race (OR = 2.516, 95% CI: 1.549~4.15, P < 0.0001), grade III (OR = 4.447, 95% CI: 2.808~7.241, P = 0.0011), total mastectomy (OR = 2.817, 95% CI: 2.092~3.822, P < 0.0001), and no surgery (OR = 7.431, 95% CI: 2.471~19.346, P < 0.0001) were independent risk factors (Table 3).

**Table 1. Clinicopathological features of training and validation sets.**

|  | All Patients | Training | Validation | P |
|---|---|---|---|---|
|  | (N = 3951) | (N = 2765) | (N = 1186) |  |
| **LNM** |  |  |  |  |
| No | 3707 (93.8%) | 2579 (93.9%) | 1110 (93.6%) | 0.744 |
| Yes | 244 (6.2%) | 168 (6.1%) | 76 (6.4%) |  |
| **Age,years** |  |  |  |  |
| Median | 59 | 59 | 59 | 0.957 |
| <50 | 889 (22.5%) | 621 (22.5%) | 268 (22.6%) |  |
| ≥50 | 3062 (77.5%) | 2144 (77.5%) | 918 (77.4%) |  |
| **Race** |  |  |  | 0.243 |
| Asian and Other | 574 (14.5%) | 406 (14.7%) | 168 (14.2%) |  |
| Black | 471 (11.9%) | 314 (11.4%) | 157 (13.2%) |  |
| White | 2906 (73.6%) | 2045 (74.0%) | 861 (72.6%) |  |
| **Grade** |  |  |  | 0.428 |
| 1 | 1170 (29.6%) | 806 (29.2%) | 364 (30.7%) |  |
| 2 | 1724 (43.6%) | 1226 (44.3%) | 498 (42.0%) |  |
| 3 | 1023 (25.9%) | 707 (25.6%) | 316 (26.6%) |  |
| 4 | 34 (0.9%) | 26 (0.9%) | 8 (0.7%) |  |
| **Laterality** |  |  |  | 0.296 |
| Left | 2047 (51.8%) | 1417 (51.2%) | 630 (53.1%) |  |
| Right | 1904 (48.2%) | 1348 (48.8%) | 556 (46.9%) |  |
| **Location** |  |  |  | 0.443 |
| Lower-inner | 262 (6.6%) | 173 (6.3%) | 89 (7.5%) |  |
| Lower-outer | 279 (7.1%) | 188 (6.8%) | 91 (7.7%) |  |
| Upper-inner | 407 (10.3%) | 282 (10.2%) | 125 (10.5%) |  |
| Upper-outer | 1396 (35.3%) | 991 (35.8%) | 405 (34.1%) |  |
| Other | 1607 (40.7%) | 1131 (40.9%) | 476 (40.1%) |  |
| **ER** |  |  |  | 0.822 |
| Nagetive | 922 (23.3%) | 642 (23.2%) | 280 (23.6%) |  |
| Positive | 3029 (76.7%) | 2123 (76.8%) | 906 (76.4%) |  |
| **PR** |  |  |  | 0.727 |
| Nagetive | 1487 (37.6%) | 1046 (37.8%) | 441 (37.2%) |  |
| Positive | 2464 (62.4%) | 1719 (62.2%) | 745 (62.8%) |  |
| **HER-2** |  |  |  | 0.831 |
| Nagetive | 2754 (69.7%) | 1924 (69.6%) | 830 (70.0%) |  |
| Positive | 1197 (30.3%) | 841 (30.4%) | 356 (30.0%) |  |
| **Surgery** |  |  |  | 0.644 |
| Breast conservation | 2329 (58.9%) | 1643 (59.4%) | 686 (57.8%) |  |
| Mastectomy | 1594 (40.3%) | 1103 (49.9%) | 491 (41.4%) |  |
| no surgery or unknown | 28 (0.7%) | 19 (0.7%) | 9 (0.8%) |  |

## Development and verification of the nomogram

The multivariate binary logistic regression analysis results identified age at diagnosis, race, grade, and surgical method as risk factors. We included these four variables and constructed a nomogram (Fig 2). The AUCs of the training set (Fig 3A) and the validation set (Fig 3B) were 0.739 (95% CI: 0.702~0.775) and 0.732 (95% CI: 0.675~0.788), respectively, indicating that the nomogram has an excellent ability to predict lymph node metastasis. In addition, the slopes of the calibration curves of the training set (Fig 3D) and the validation set (Fig 3E) were close to 1,

**Table 2.  The clinicopathological characteristics in the SJTU-BCDB.**

| | Test cohort |
|---|---|
| | (N = 323) |
| **LNM** | |
| No | 309 (95.6%) |
| Yes | 14 (4.3%) |
| **Age,years** | |
| <50 | 129 (39.9%) |
| ≥50 | 194 (60.1%) |
| **Race** | |
| Asian and Other | 323 (100%) |
| Black | 0 (0%) |
| White | 0 (0%) |
| **Grade** | |
| 1 | 7 (2.1%) |
| 2 | 92 (28.4%) |
| 3 | 224 (69.3%) |
| 4 | 0 (0%) |
| **Laterality** | |
| Left | 170 (52.6%) |
| Right | 153 (47.4%) |
| **Location** | |
| Lower-inner | 17 (5.2%) |
| Lower-outer | 38 (11.7%) |
| Upper-inner | 48 (14.8%) |
| Upper-outer | 112 (34.6%) |
| Other | 108(33.4%) |
| **ER** | |
| Nagetive | 177 (54.8%) |
| Positive | 146 (45.2%) |
| **PR** | |
| Nagetive | 214 (66.3%) |
| Positive | 109 (33.7%) |
| **HER-2** | |
| Nagetive | 153(47.4%) |
| Positive | 170 (52.6%) |
| **Surgery** | |
| Breast conservation | 64 (19.8%) |
| Mastectomy | 259 (80.2%) |
| no surgery or unknown | 0 (0%) |

and the P values of the Hosmer–Lemeshow test were 0.691 and 0.353, showed that the predicted values of the prediction model were in good agreement with the actual values. The evaluation effect of the external test cohort is essentially the same as the validation cohort (Fig 3C and 3F).

## Model benefit

The clinical benefit of the nomogram was evaluated by decision curve analysis (DCA). When the nomogram was used to asses the patient's probability of LNM reaching the risk threshold probability, DCA results can be used to help initiate early intervention to benefit the patient.

**Table 3. Risk variables for LNM determined by univariate and multivariate logistic regression analyses.**

| | Univariate analysis | | | Multivariate analysis | | |
|---|---|---|---|---|---|---|
| | OR | 95%CI | P | OR | 95%CI | P |
| **Age,years** | | | **<0.0001** | | | **<0.0001** |
| <50 | Reference | | | Reference | | |
| ≥50 | 0.377 | 0.288–0.494 | | 0.537 | 0.403–0.719 | |
| **Race** | | | **<0.0001** | | | **0.0019** |
| Asian and Other | Reference | | | Reference | | |
| Black | 2.299 | 1.443–3.723 | | 2.516 | 1.549–4.150 | |
| White | 1.062 | 0.721–1.616 | | 1.240 | 0.833–1.907 | |
| **Grade** | | | **<0.0001** | | | **0.0011** |
| I | Reference | | | Reference | | |
| II | 2.470 | 1.626–3.879 | | 2.415 | 1.568–3.838 | |
| III | 5.210 | 3.445–8.159 | | 4.447 | 2.808–7.241 | |
| IV | 1.658 | 0.168–6.806 | | 1.678 | 0.166–7.278 | |
| **Laterality** | | | **<0.0001** | | | **0.231** |
| Left | Reference | | | Reference | | |
| Right | 1.191 | 0.916-.548 | | 1.218 | 0.928–1.600 | |
| **Location** | | | **<0.0001** | | | **0.099** |
| Lower-inner | Reference | | | Reference | | |
| Lower-outer | 2.413 | 1.104–5.752 | | 2.301 | 1.033–5.572 | |
| Upper-inner | 1.237 | 0.548–3.002 | | 1.170 | 0.509–2.883 | |
| Upper-outer | 1.985 | 1.029–4.377 | | 1.903 | 0.971–4.241 | |
| Others | 1.808 | 0.939–3.981 | | 1.505 | 0.769–3.349 | |
| **ER** | | | **<0.0001** | | | **0.101** |
| Negative | Reference | | | Reference | | |
| Positive | 0.765 | 0.573–1.031 | | 1.487 | 0.996–2.215 | |
| **PR** | | | **<0.0001** | | | **0.721** |
| Negative | Reference | | | Reference | | |
| Positive | 0.721 | 0.554–0.940 | | 0.924 | 0.646–1.336 | |
| **HER2** | | | **<0.0001** | | | **0.841** |
| Negative | Reference | | | Reference | | |
| Positive | 1.647 | 1.258–2.149 | | 0.962 | 0.701–1.315 | |
| **Surgery** | | | **<0.0001** | | | **<0.0001** |
| Breast conservation | Reference | | | Reference | | |
| Mastectomy | 3.356 | 2.540–4.473 | | 2.817 | 2.092–3.822 | |
| no surgery or unknown | 8.000 | 2.796–19.525 | | 7.431 | 2.471–19.346 | |

Fig 4 shows that the nomogram can maximize the net benefit compared with the prediction of LNM by age at diagnosis, race, grade, or surgical method alone, suggesting that this prediction model has good clinical efficacy.

## Discussion

DCIS-MI is a particular type of breast cancer between ductal carcinoma in situ (DCIS) and invasive ductal carcinoma (IDC), with an infiltration range of less than 1 mm. Studies have shown that DCIS, DCIS-MI, and IDC have the same gene expression [10]. The three may be a continuous process, and DCIS-MI is a transitional stage from DCIS to IDC [11, 12]. However, recent studies suggest differences in the expression of certain molecules among the three, and

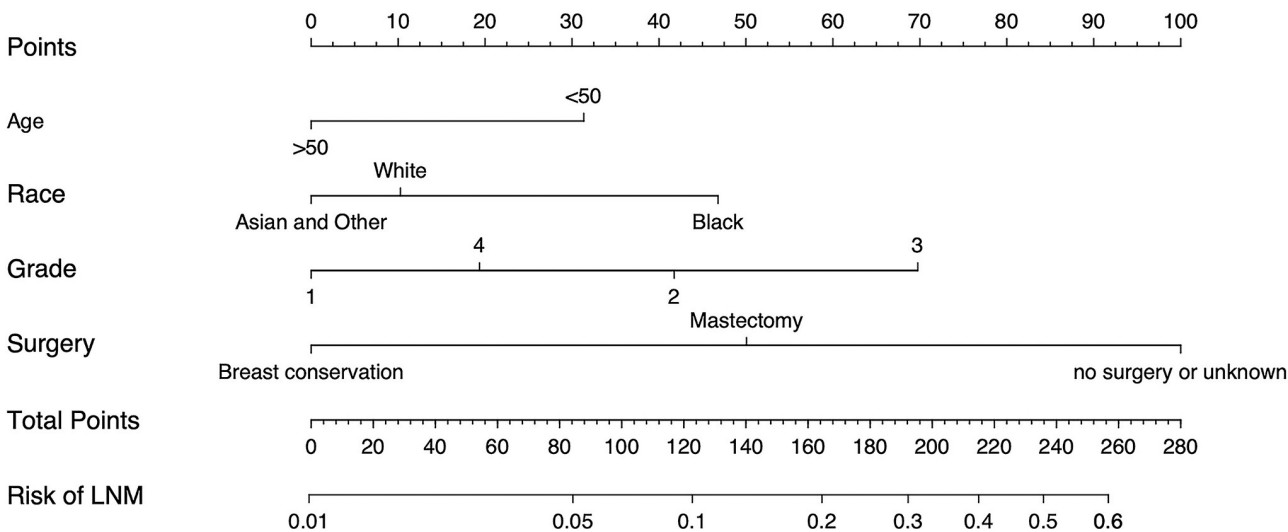

**Fig 2. Nomogram predicting regional LNM in patients with DCIS-MI based on training set.** Note: The first row is the point assignment for each variable. Rows 2–5 indicate the variables included in the nomogram. For an individual patient, each variable is assigned a point value based on the histopathological characteristics. The points for each variable were summed and located on the total point line. And then, the bottom line shows the probability of the patient having regional LNM.

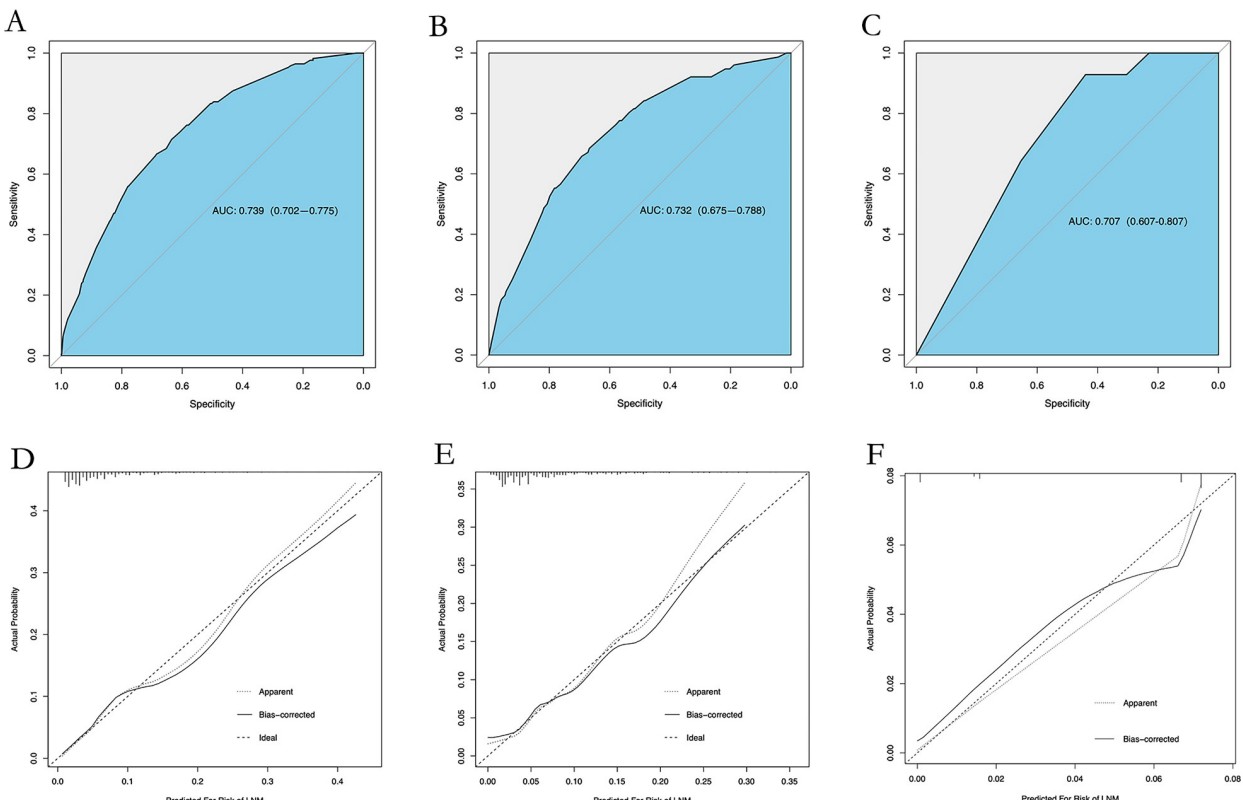

**Fig 3. Discrimination and validation of nomogram for predicting LNM in patients with DCIS-MI.** The ROC curves of the nomogram for predicting LNM in the training cohort (A), validation cohort (B) and test cohort (C). Calibration plots for the nomogram in the training cohort (D), validation cohort (E) and test cohort (F). (AUC, area under the curve; 95%CI, 95% confidence interval).

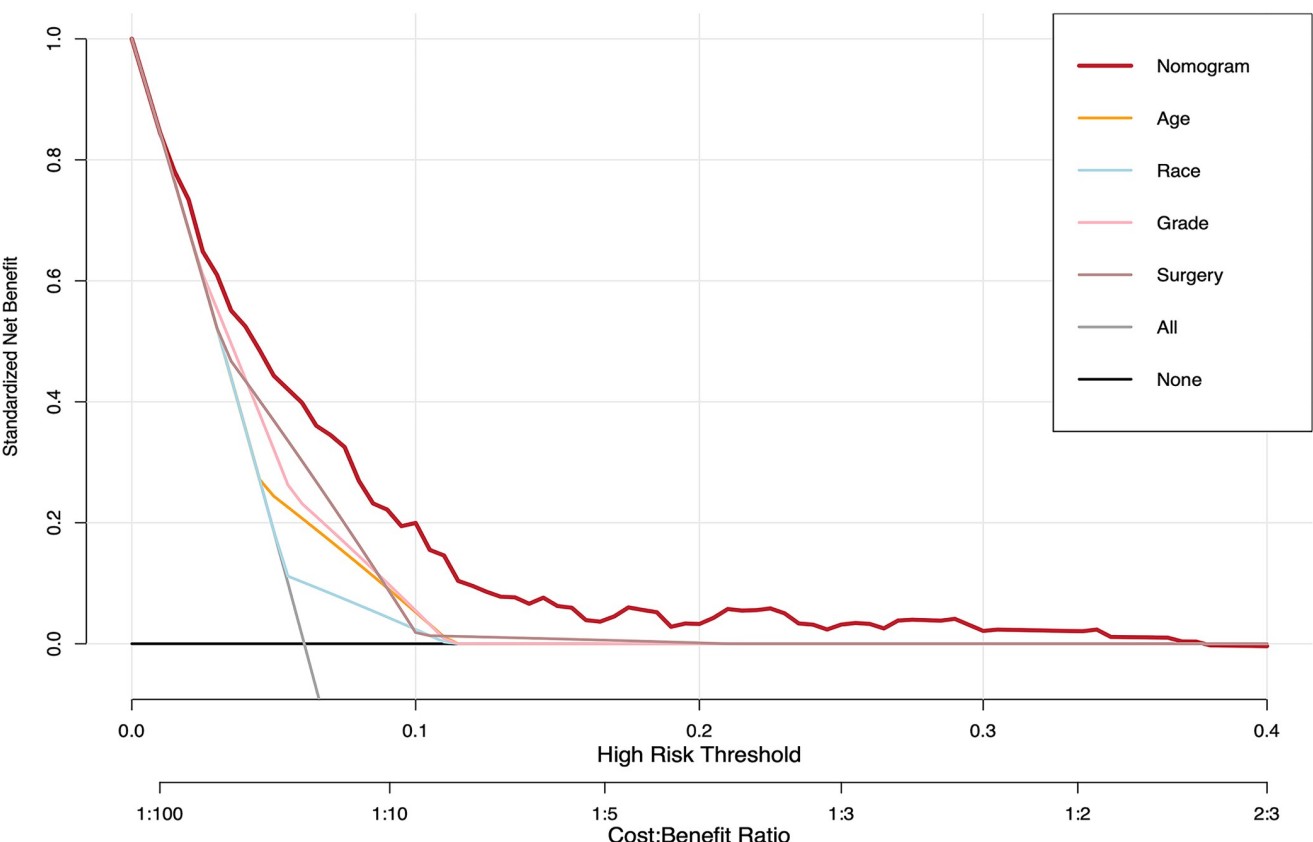

**Fig 4. Decision curve for prediction of regional LNM for DCIS-MI.** Black line: assume no patient will have regional LNM; gray line: assume all patients will have regional LNM; orange line: binary decision rule based on age alone; blue line: binary decision rule based on race alone; pink line: binary decision rule based on grade alone; brown line: binary decision rule based on surgery alone; red line: decision based on nomogram. The x-axis and the y-axis were the threshold probability and the net benefit, respectively.

DCIS-MI can be regarded as an independent tumor with potential metastasis and invasion [13–15]. The lymph node metastasis rate of DCIS-MI is currently believed to be 0%-20% [16–21]. In this study, 4274 patients with DCIS-MI were included, of whom 258 had lymph node metastasis, and the metastasis rate was 6.03%, consistent with previous research results.

The diagnosis of DCIS-MI usually depends on postoperative routine pathological results. However, studies have shown that the sensitivity and specificity of intraoperative frozen section results were 45.8% and 64.7%, respectively [22]. Schueller et al reported 1352 cases of ultrasound-guided core needle biopsy, with a false-negative rate of 1.6% and an underestimation rate of 31.4% [8], but there was still missed diagnosis. Therefore, reoperation is usually required when DCIS-MI is not found before or during surgery and is diagnosed after surgery. Breast-conserving surgery or mastectomy is generally selected, and lymph node treatment is controversial. LNM is considered an important adverse prognostic factor, and the 5-year survival rate of lymph node-positive DCIS-MI patients is significantly lower than that of lymph node-negative DCIS-MI patients [5]. Therefore, lymph node management is crucial for the treatment decision of DCIS-MI patients.

In this study, based on a large number of cases in the SEER database, four risk factors for age, race, grade, and surgical type were determined by univariate and multivariate logistic

regression analysis. A prediction model was constructed and verified by ROC, DCA, and calibration curves. The AUC of the training set, the validation set, and the external test set were greater than 0.7, and the slope of the calibration curve was close to 1, indicating that the prediction model had good discrimination and calibration [23, 24]. At the same time, the DCA curve shows that this model has greater clinical benefits. Therefore, the combined model of this study may be applied to the prediction of LNM in clinical microinvasive breast cancer.

In our study, age, race, grade, and surgical procedure were found to be essential factors in predicting regional lymph node metastasis in patients with DCIS-MI. Previous studies have shown that the median age of diagnosis of DCIS-MI patients with LNM is 54 [25]. The median age of diagnosis of this group of patients in this study cohort is 59 years old, and all are older than 50 years old. In addition, patients younger than 50 have a higher risk of LNM. At the same time, we found that black people are more likely to have LNM than Asians and white people, consistent with previous studies. In addition, several studies have shown that regional lymph node status is closely related to grade, and the higher the pathological grade is, the higher the positive rate of regional lymph nodes [26, 27]. A higher histological grade means a lower degree of tumor differentiation and generally a higher degree of malignancy. However, this study showed that the risk of grade III was the highest, and the risk of grade IV was only higher than that of grade I. Previous studies have not found this result. Due to the small number of patients with grade IV (only 34 patients), there may be data bias, and more results are needed for verification. This study also found a correlation between surgical methods and LNM. The risk of lymph node metastasis in patients undergoing total mastectomy was higher than that of breast-conserving surgery, suggesting that lymph node metastasis was closely related to the size of the lesion (DCIS) and the multifocal lesions because patients with larger masses and multifocal lesions were often not suitable for breast-conserving surgery. However, the size of the primary tumor and the number of infiltrating lesions in patients with DCIS-MI cannot be obtained from the SEER database. Finally, for example, ER and other variables have been found to be associated with LNM, but our study did not show a correlation with LNM.

There have been many reports on the analysis of risk factors for lymph node metastasis in microinvasive breast cancer, but there are few studies using similar risk factors to construct a nomogram. Ko et al. used 9635 DCIS-MI patients from Asan Medical Center to analyze the risk factors for axillary lymph node metastasis and identified ER-positive status and lymphatic invasion as independent predictors [28]. Chen et al. conducted 1:1 propensity score matching on patient data to eliminate potential bias and found that young age and high-grade lesions were independent risk factors [29]. Gooch et al. used the independent risk factors obtained from the analysis to construct a nomogram for predicting the occurrence of LNM in DCIS-MI, with a c-index of 0.71, but did not verify the calibration and clinical effectiveness of the model [25]. In our study, we constructed a predictive model for lymph node metastasis in patients with DCIS-MI and included the surgical procedure as a variable for the first time. At the same time, we effectively verified the constructed prediction model. The calibration and DCA curves confirmed that the model had good calibration and clinical benefits.

Although we have constructed an accurate and practical prediction model, some things could be improved. First, because the data of this study were obtained from the SEER database, some potentially influential factors were not included, such as nerve and vascular invasion, Ki-67, and other clinicopathological indicators. Secondly, this study is a retrospective study of big data, and there may be bias in data screening. For example, we excluded some patients with unknown information. Finally, although the external cohort from SJTU-BCDB may help avoid overfitting of the model, the data in the external validation set may have been insufficient and more cases from Eastern countries are needed.

## Conclusion

In conclusion, through logistic analysis, we found that age at diagnosis, race, grade, and surgical procedure were risk factors for LNM in patients with DCIS-MI and provided an accurate, practical, and effective lymph node metastasis prediction model for DCIS-MI patients. By using four simple and easily available variables, the possibility of LNM can be predicted more accurately. This has good clinical value for the management of axillary lymph nodes in such patients.

## Supporting information

**S1 Data.**
(XLSX)

**S2 Data.**
(XLSX)

**S3 Data.**
(XLSX)

**S4 Data.**
(XLSX)

## Acknowledgments

The authors thank all patients, investigators and institutions involved in these studies, especially the SEER and SJTU-BCDB databases.

## Author Contributions

**Conceptualization:** Yuan Sui, Liwei Meng.

**Data curation:** Kaijun Zhu, Yuan Sui.

**Formal analysis:** Kaijun Zhu, Mingliao Zhu.

**Funding acquisition:** Liwei Meng.

**Investigation:** Yuan Sui, Jiangfeng Dai, Zhian Li.

**Methodology:** Kaijun Zhu, Yuan Gao, Ying Yuan, Pujian Sun.

**Project administration:** Liwei Meng, Zhian Li.

**Resources:** Liwei Meng, Zhian Li.

**Software:** Yuan Sui.

**Supervision:** Yuan Gao, Liwei Meng.

**Validation:** Mingliao Zhu, Ying Yuan, Pujian Sun.

**Visualization:** Kaijun Zhu, Yuan Sui, Ying Yuan, Pujian Sun.

**Writing – original draft:** Kaijun Zhu, Yuan Sui.

**Writing – review & editing:** Mingliao Zhu, Yuan Gao, Liwei Meng.

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
