## [Decision Letter · Decision Letter 0]

26 Feb 2024

PONE-D-23-35296Development and validation of a nomogram for predicting lymph node metastasis in ductal carcinoma in situ with microinvasion: A SEER population-based studyPLOS ONE

Dear Dr. Meng,

Thank you for submitting your manuscript to PLOS ONE. After careful consideration, we feel that it has merit but does not fully meet PLOS ONE’s publication criteria as it currently stands. Therefore, we invite you to submit a revised version of the manuscript that addresses the points raised during the review process.

We sincerely apologise for the delays in this process. As conflicting responses were received, we needed to engage a third reviewer.

We look forward to receiving your revised manuscript.

Kind regards,

Amy McCart Reed

Academic Editor

PLOS ONE

Journal Requirements:

3. Thank you for stating the following financial disclosure:"This work was supported by Nonprofit Science and Technology Projects of Zhejiang Medical and Health (No. 2021KY369). "  

Reviewers' comments:

Reviewer's Responses to Questions

**Comments to the Author**

1. Is the manuscript technically sound, and do the data support the conclusions?

Reviewer #1: Yes

Reviewer #2: Yes

Reviewer #3: Yes

2. Has the statistical analysis been performed appropriately and rigorously? 

Reviewer #1: Yes

Reviewer #2: Yes

Reviewer #3: Yes

3. Have the authors made all data underlying the findings in their manuscript fully available?

Reviewer #1: Yes

Reviewer #2: Yes

Reviewer #3: Yes

4. Is the manuscript presented in an intelligible fashion and written in standard English?

Reviewer #1: No

Reviewer #2: Yes

Reviewer #3: Yes

5. Review Comments to the Author

Reviewer #1: I would like to congratulate the authors of the article "Development and validation of a nomogram for predicting lymph node metastasis in ductal carcinoma in situ with microinvasion: A SEER population-based study" for the originality and quality of the data presented.

As suggestions

1. Review English

2. Place captions on the tables, especially abbreviations.

Reviewer #2: Dr. Zhu explored a nomogram for predicting LNM in DCISM using SEER database and validated in an external database from Shanghai Jiao Tong University.

My major concerns are as follow.

1. Prognostic factors for DCISM has been discussed elsewhere(PMID: 34547183; PMID: 31529311; PMID: 30588019). Therefore the novelty of this manuscript is not good.

2. The quality of the images are not good, the figure legend is blur.

3. Since race is one of the risk factors how do the authors validate race in Shanghai Jiao Tong University Breast Cancer Database (SJTU-BCDB)

Reviewer #3: The authors developed and validated a predictive model for lymph node metastasis (LNM) in ductal carcinoma in situ (DCIS) using data from the SEER database and an independent dataset from a cohort in Shanghai. I have questions about the variable selections in the model.

Method of treatment was used as a predictor but the decision of treatment will be influenced by clinical manifestations and the doctors will most likely already taken the probability of LNM into account when deciding on treatment plan. Using this as a predictor for LNM seems to be a bit circular? How would the model perform if this variable is not included in the prediction model?

Didn’t patients with “no surgery or unknown” got excluded from the analysis as specified in lines 75-76? How was LNM determined in these patients? Lymph node biopsy?

Table 2. Need to list LNM status. Based on Figure 3 there seemed to be only a few LNM events in the Shanghai data. A larger data set as an external validation data would be more desirable.

Age was used as a categorical variable (<50 or >=50) in the model. Was that because continuous age data were not available in the external validation data? Will include age as a continuous variable in the model improve the model performance?

6. PLOS authors have the option to publish the peer review history of their article (what does this mean?). If published, this will include your full peer review and any attached files.

Reviewer #1: No

Reviewer #2: No

Reviewer #3: No

---

## [Author Response · Author response to Decision Letter 0]

5 Mar 2024

Reviewer #1: I would like to congratulate the authors of the article Development and validation of a nomogram for predicting lymph node metastasis in ductal carcinoma in situ with microinvasion: A SEER population-based study; for the originality and quality of the data presented.

As suggestions

1. Review English

RESPONSE: We thank the reviewer for this valuable suggestion, and the whole manuscript has been polished accordingly by Elsevier.

2. Place captions on the tables, especially abbreviations.

RESPONSE: Thank you for your careful examination, we are sorry for our carelessness. According to your opinion, We have made corrections, the specific changes are highlighted in yellow in the Revised Manuscript with Track Changes.

Reviewer #2: Dr. Zhu explored a nomogram for predicting LNM in DCISM using SEER database and validated it in an external database from Shanghai Jiao Tong University. My major concerns are as follow.

1. Prognostic factors for DCISM has been discussed elsewhere(PMID: 34547183; PMID: 31529311; PMID: 30588019). Therefore the novelty of this manuscript is not good.

RESPONSE: Thank you for your feedback and raising the concern regarding the novelty of our study. We appreciate your attention to previously published studies in the field. We would like to address your comment and explain why our study still contributes to the existing knowledge and advances the field. First of all, our study not only discussed the risk factors of lymph node metastasis in DCIS-MI but also constructed a predictive model for lymph node metastasis and verified it. However, the two articles you mentioned (PMID: 34547183, PMID: 30588019) did not construct a predictive model. Although PMID: 31529311 also constructed a related model, they did not verify the model. We have reason to question the practicality of the model they constructed, and our research not only increased the Decision curve. The validation of external data sets from China is also included to ensure the validity of the model. We hope this clarification addresses your concerns regarding the novelty of our study. We appreciate your time and consideration, and we remain open to any further suggestions or feedback you may have.

2. The quality of the images are not good, the figure legend is blur.

RESPONSE: Thanks for this feedback. According to your suggestion, we improved the quality of the image, and the images we uploaded fully meet the publishing requirements of PLOS ONE magazine, For the related images with errors, we have made modifications.

3. Since race is one of the risk factors how do the authors validate race in Shanghai Jiao Tong University Breast Cancer Database (SJTU-BCDB)

RESPONSE: We put the constructed model into the external validation set for testing and found that the accuracy and authenticity of the model had good results. The data set from SJTU-BCDB is only used as the external validation of the model, not the validation of risk factors. Therefore, we believe that this is a good prediction model and is applicable to the Chinese population. At the same time, we have to admit that due to the temporary inability to obtain multi-ethnic external data, this is a disadvantage of our research.

Reviewer #3: The authors developed and validated a predictive model for lymph node metastasis (LNM) in ductal carcinoma in situ (DCIS) using data from the SEER database and an independent dataset from a cohort in Shanghai. I have questions about the variable selections in the model.

1.Method of treatment was used as a predictor but the decision of treatment will be influenced by clinical manifestations and the doctors will most likely already taken the probability of LNM into account when deciding on treatment plan. Using this as a predictor for LNM seems to be a bit circular? How would the model perform if this variable is not included in the prediction model?

RESPONSE: We sincerely thank you for your careful reading. The classification of surgical methods included in the study included breast-conserving surgery and mastectomy. According to the latest NCCN guidelines, clinicians ' choice of two surgical methods is not affected by lymph node metastasis. Due to the lack of DCIS-MI-related variables in the SEER database, including the size of the focus, the existence of multiple infiltrations, etc. Therefore, we included the surgical methods as a new variable. Because the choice of surgical methods has taken into account the size of the focus, multifocal, etc. We compared the two prediction models with or without the variable of surgical methods. When this variable is not included, the discrimination of the prediction model will decrease.

2.Didn’t patients with no surgery or unknown get excluded from the analysis as specified in lines 75-76? How was LNM determined in these patients? Lymph node biopsy?

RESPONSE: We are sorry for our careless mistakes. Thank you for your reminder. This study did not exclude patients with no surgery or unknown. Because we believe that this choice was comprehensively considered by clinicians and could be included in the predictive variables. Since these data come from the SEER database, after careful review, we cannot confirm how the patient 's lymph node metastasis is detected. The SEER database only gives the results of lymph node metastasis.

3.Table 2. Need to list LNM status. Based on Figure 3 there seemed to be only a few LNM events in the Shanghai data. A larger data set as an external validation data would be more desirable.

RESPONSE: Relevant modifications have been marked in the Revised Manuscript with Track Changes document. The external test set included 14 patients with lymph node metastasis, accounting for 4.3 %. This data is similar to the proportion of the SEER library and similar to other research results. Due to the low incidence of DCIS-MI and the rare occurrence of lymph node metastasis, we believe that the external test set of this study is appropriate. We admit that it is better if there is a larger data set.

4.Age was used as a categorical variable (<50 or >=50) in the model. Was that because continuous age data were not available in the external validation data? Will include age as a continuous variable in the model improve the model performance?

RESPONSE: This study is based on binary logistic regression analysis of risk factors. Therefore, compared with continuous variables, the definition of diagnostic age as a categorical variable is more conducive to the construction of the model and can improve the accuracy of the model.

---

## [Editor Report · Decision Letter 1]

11 Mar 2024

Development and validation of a nomogram for predicting lymph node metastasis in ductal carcinoma in situ with microinvasion: A SEER population-based study

PONE-D-23-35296R1

Dear Dr. Meng,

We’re pleased to inform you that your manuscript has been judged scientifically suitable for publication and will be formally accepted for publication once it meets all outstanding technical requirements.

Kind regards,

Amy McCart Reed

Academic Editor

PLOS ONE
---

## [Editor Report · Acceptance letter]

22 Mar 2024

PONE-D-23-35296R1 

PLOS ONE

Dear Dr. Meng, 

I'm pleased to inform you that your manuscript has been deemed suitable for publication in PLOS ONE. Congratulations! Your manuscript is now being handed over to our production team.

Kind regards, 

on behalf of

Associate Professor Amy McCart Reed 

Academic Editor

PLOS ONE